# Sequences within and upstream of the mouse Ets1 gene drive high level expression in B cells, but are not sufficient for consistent expression in T cells

**Alyssa Kearly[1], Prontip Saelee[1], Jonathan Bard[1], Satrajit Sinha[1], Anne Satterthwaite[2], Lee Ann Garrett-Sinha**[1]*

1 Department of Biochemistry, Jacobs School of Medicine and Biomedical Sciences, State University of New York at Buffalo, Buffalo, New York, United States of America, 2 Department of Internal Medicine, University of Texas Southwestern Medical Center, Dallas, Texas, United States of America

* leesinha@buffalo.edu

## Abstract

The levels of transcription factor Ets1 are high in resting B and T cells, but are downregulated by signaling through antigen receptors and Toll-like receptors (TLRs). Loss of Ets1 in mice leads to excessive immune cell activation and development of an autoimmune syndrome and reduced Ets1 expression has been observed in human PBMCs in the context of autoimmune diseases. In B cells, Ets1 serves to prevent premature activation and differentiation to antibody-secreting cells. Given these important roles for Ets1 in the immune response, stringent control of *Ets1* gene expression levels is required for homeostasis. However, the genetic regulatory elements that control expression of the *Ets1* gene remain relatively unknown. Here we identify a topologically-associating domain (TAD) in the chromatin of B cells that includes the mouse *Ets1* gene locus and describe an interaction hub that extends over 100 kb upstream and into the gene body. Additionally, we compile epigenetic datasets to find several putative regulatory elements within the interaction hub by identifying regions of high DNA accessibility and enrichment of active enhancer histone marks. Using reporter constructs, we determine that DNA sequences within this interaction hub are sufficient to direct reporter gene expression in lymphoid tissues of transgenic mice. Further analysis indicates that the reporter construct drives faithful expression of the reporter gene in mouse B cells, but variegated expression in T cells, suggesting the existence of T cell regulatory elements outside this region. To investigate how the downregulation of Ets1 transcription is associated with alterations in the epigenetic landscape of stimulated B cells, we performed ATAC-seq in resting and BCR-stimulated primary B cells and identified four regions within and upstream of the *Ets1* locus that undergo changes in chromatin accessibility that correlate to *Ets1* gene expression. Interestingly, functional analysis of several putative Ets1 regulatory elements using luciferase constructs suggested a high level of functional redundancy. Taken together our studies reveal a complex network of regulatory elements and transcription factors that coordinate the B cell-specific expression of *Ets1*.

**Data availability statement:** The ATAC-seq datasets for unstimulated B cells and BCR-stimulated B cells collected for this study are publicly available in NCBI's Gene Expression Omnibus (GEO) under accession number GSE273513. All other ATAC-seq, ChIP-seq and Hi-Seq datasets analyzed in this manuscript were data collected by other research groups and are all available in the GEO database under the following accession numbers: GSE82144 [32]; GSE29184 [34]; GSE60103 [35]; GSE36030 [36], GSE100738 [37]; GSE109125 [37], GSE29611 [39]; GSE18927 [40]; GSE51334; and GSE118253 [41].

**Funding:** The funding for this project was from grant R01 AI122720 funded by the National Institutes of Allergy and Infectious Disease (NIAID). The funders had no role in study design, data collection and analysis, decision to publish, or preparation of the manuscript.

**Competing interests:** The authors have declared that no competing interests exist.

## Introduction

Autoimmune diseases are caused by the immune system attacking self-tissues. Genetic factors influence the development of autoimmune diseases and genome-wide association studies (GWAS) have been used to identify disease-associated single-nucleotide polymorphisms (SNPs) for many autoimmune diseases. GWAS has established that the human *ETS1* gene is a susceptibility locus for multiple autoimmune diseases, such as systemic lupus erythematosus (SLE), rheumatoid arthritis (RA), psoriasis and others [1]. Ets1 mRNA levels are reduced in PBMCs isolated from patients with a variety of different autoimmune diseases [2–7]. Supporting a role for Ets1 in autoimmunity, mice lacking Ets1 develop a lupus-like autoimmune disease, with excessive B and T cell activation and secretion of IgM and IgG autoantibodies against self-antigens [8–11]. Cell-type-specific deletion of Ets1 has demonstrated cell-intrinsic roles for Ets1 in blocking the aberrant activation of both B and T cells [10,11].

Ets1 expression is high in resting lymphocytes [12–14], but is downregulated by stimulation through antigen receptors or Toll-like receptors [15–19]. Loss of Ets1 leads to increased B cell differentiation into antibody-secreting plasma cells [20] and forced expression of Ets1 blocks development of plasma cells [7,15,16,21]. Since the level of *Ets1* in B cells is crucial to regulating the differentiation program, expression of the *Ets1* gene must be tightly controlled. Maintenance of Ets1 expression in resting B cells relies heavily on inhibitory signaling pathways that limit the transduction of activating signals. B cells that lack expression of inhibitory signaling components such as inhibitory receptors CD22 or SiglecG, proximal kinase Lyn, or downstream phosphatases SHP-1 or SHIP-1 have increased BCR signaling and express lower levels of Ets1 mRNA and protein [16]. Our studies have shown that the low level of Ets1 in Lyn-/- B cells is not due to decreased stability of the *Ets1* protein or mRNA [16]. Instead, BCR and TLR signaling result in a decrease in transcription of the *Ets1* gene and this does not require new protein synthesis, implicating pre-formed transcription factors [17].

Transcriptional regulation of gene expression is controlled by three-dimensional chromatin architecture in the nucleus as well as the binding of cell-type specific transcription factors. As part of this process, the genome is partitioned into an array of chromatin loops generated by the combined actions of cohesin and CTCF proteins that define topologically-associated domains (TADs) [22–24]. TADs counteract the formation of transcriptionally-silent heterochromatin domains [25,26] and also function to insulate promoters of genes within them from the influence of response elements outside the TAD [27–30]. TADs are further sub-divided into sub-TAD regions that contain one or a few genes along with regulatory elements such as gene enhancers or silencers that bind specific transcription factors [26–28]. Recently, the organization TAD structures in the *Ets1* locus in mouse and human T cells have been published [31]. This analysis also identified a super-enhancer region located approximately 250 kb downstream of the *Ets1* gene, which is required for CD4+ T helper 1 (Th1) differentiation [31]. In contrast to the situation in T cells, the genomic organization of the *Ets1* locus and the regulatory elements and transcription factors that bind to them in B cells remain unknown.

To identify B cell TADs and sub-TADs around *Ets1* and also regulatory elements that coordinate expression of Ets1, we examined chromatin interactions, epigenetic profiles and DNA accessibility in mouse B cells. To further validate the roles of the regulatory elements, we generated a bacterial artificial chromosome (BAC) reporter construct in which eGFP expression was inserted under the direction of sequences from -113 kb to +105 kb of the mouse *Ets1* gene and observed reporter expression that recapitulated endogenous *Ets1* in B cells, but did not fully recapitulate *Ets1* expression in T cells. Four putative regulatory elements experience

changes in accessibility upon BCR stimulation and motif analysis identified transcription factors that may bind to these regulatory elements. Transient transfection assays with numerous putative regulatory element sequences failed to detect sequences that individually activate or repress reporter gene transcription at an appreciable level. These results suggest the possibility that *Ets1* gene transcription in B cells may be regulated by a network of multiple regulatory elements the coordinately stimulate gene transcription, rather than being reliant on one or a few dominant regulatory elements.

## Materials and methods

### Analysis of publicly-available Hi-C, ChIP-seq and ATAC-seq datasets

Hi-C data (GSE82144) [32] from resting mouse splenic B cells was visualized using the Juicebox desktop application [33]. ChIP-seq datasets analyzed include: CTCF ChIP-seq of mouse spleen cells (GSE29184) [34]; H3K4me1, H3K4me3 and H3K27ac ChIP-seq of FACS-sorted (CD3-, B220 + , CD19+) mouse splenic B cells (GSE60103) [35]; p300 ChIP-seq of resting mouse splenic B cells (GSE82144) [32]; and c-Jun and JunD ChIP-seq of mouse the CH12 B lymphoma cell line (GSE36030) [36]. ATAC-seq and RNA-seq datasets were retrieved from the ImmGen data series (GSE100738, ATAC-Seq and GSE109125, RNA-Seq) with the sorted immune cell populations [37]. The ImmGen cell populations analyzed were: bone marrow hematopoietic stem cells (HSC), bone marrow multi-potent progenitors (MPP), bone marrow common lymphoid progenitors (CLP), bone marrow pro-B cells (pro-B), bone marrow immature B cells (Imm B), total splenic B cells, splenic follicular B cells, splenic marginal zone B cells, splenic germinal center B cells, splenic plasma cells, naïve splenic CD4 + T cells, naïve splenic CD8 + T cells, splenic NK cells, splenic macrophages, splenic neutrophils and splenic dendritic cells (DC). Sorting parameters for these populations are listed in Yoshida et al [37]. In addition, ATAC-seq data from FACS-sorted (CD3-, B220 + , CD19+) mouse splenic B cells was also used (GSE60103) [35]. The JASPAR 2020 database of transcription factor binding sites was used to identify transcription factor motifs [38].

For human cells, the following datasets were analyzed: CD20 + B cell CTCF ChIP-seq (GSE29611) [39]; CD19 + B cell H3K27ac ChIP-seq (GSE18927) [40]; GM12878 B lymphoma H3K27ac ChIP-seq (GSE51334) and naïve B cell ATAC-seq (GSE118253) [41]. All of the datasets were visualized using the UCSC Genome Browser [42].

### BAC recombineering and transgenic mouse generation

The RP23-350A20 BAC (BACPAC Resources) was recombineered according to the protocol previous described [43] using a targeting construct that contained eGFP, an SV40 polyA sequence and a neomycin/kanamycin resistance cassette between homology arms to replace the first exon of the *Ets1* gene. Clones were selected and PCR-tested for eGFP insertion. FLP-FRT was induced to remove the neomycin/kanamycin resistance cassette. The final recombined BAC contains the eGFP cDNA and SV40 polyA sequence inserted in place of the first exon of the major isoform of Ets1 (mouse Ets1 isoform 1) and under the control of Ets1 gene regulatory elements.

The recombined BAC was microinjected into fertilized C57BL/6 mouse eggs by the Roswell Park Comprehensive Cancer Center (RPCCC) Gene Targeting and Transgenic Facility. BAC Transgenic (BACtg) mice were identified by PCR genotyping. All mice were house at RPCCC Lab Animal Shared Resource and experiments were performed in accordance with Institutional Animal Care and Use Committee protocol number UB1104M. Mice were euthanized by $CO_2$ inhalation followed by cervical dislocation.

## BACtg flow cytometry

Single-cell suspensions were made from the spleens of wild type and BACtg animals and stained with Ghost Dye Violet 510 (Tonbo Biosciences) and fluorescent dye-conjugated antibodies against B lineage surface markers, including B220, CD19, CD21, CD23, CD80, PDL2, Fas, PNA, CD138, and CD98, or T lineage surface markers, including CD3, CD4, and CD8. Cells were fixed and permeabilized for intracellular staining and stained with unconjugated rabbit monoclonal anti-Ets1 (D8O8A, Cell Signaling Technologies) or rabbit monoclonal IgG isotype control, followed by PE-conjugated anti-rabbit IgG. Data was collected using a LSR II flow cytometer and analyzed using FlowJo software. Samples were gated on singlets and live cells, and specific populations were identified based on marker expression.

## BACtg B cell isolation, stimulation, and qPCR

Splenic B cells from BACtg and wild type littermate control mice were isolated using the EasySep Mouse B Cell Isolation Kit (StemCell Technologies) and rested for 30 minutes and then either left unstimulated or stimulated with the addition of 10 µg/ml goat anti-IgM F(ab')$_2$ crosslinking antibody (Jackson ImmunoResearch) for two hours before RNA isolation. Primers for qPCR include Ets1 mRNA (F-AGTCTTGTCAGTCCTTTATCAGC, R-TTTTCCTCTTTCCCCATCTCC); Ets1-pre-mRNA Ets1 pre-mRNA set A (F-TCGATCT CAAGCCGACTCTC, R-GTCTTGGGCCACCAACAGTC); eGFP (F-CCATCTTCTTCAAG GACGAC, R-GCCATGATATAGACGTTGTGG); and beta-actin (F-GCAGCTCCTTC GTTGCCGGTC, R-TTTGCACATGCCGGAGCCGTTG).

## Mouse B cell ATAC-seq and identification of DARs

B cells were isolated from wild type mouse spleens and stimulated via the BCR for two hours. 100,000 cells from each sample were pelleted and stored frozen prior to ATAC-seq (Active Motif). Illumina NextSeq500 sequencing was used to generate paired-end 42 bp sequencing reads that were mapped to the mm10 genome build using the BWA algorithm with default settings [44]. Duplicate reads were removed and analysis was performed on reads that passed Illumina's purity filter, aligned with a maximum of 2 mismatches, and mapped uniquely. Data was normalized across samples by reducing read count to that of the smallest sample. Peaks were called using the MACS2 algorithm [45]. DARs were identified by comparing the number of sequencing tags in each region and identifying regions with greater than 2-fold differences in tag count. Gene ontology analysis of genes within 10 kb of DARs was performed using DAVID [46], limited to Gene Ontology (GO)'s biological process annotation set. *De novo* motif analysis was performed using HOMER's findMotifsGenome.pl program [47]. The FIMO program within the MEME Suite of tools was used to scan the four DARs for transcription factor motifs [48]. ATAC-seq data are available in the GEO database under the accession number GSE273513.

## Luciferase construct design and luciferase assays

A region of the mouse Ets1 proximal promoter of 469 bp (-451 to +17 bp) was cloned into the pGL3-Basic firefly luciferase vector. Sequences covering regions of Sites 1-6 (defined in the Results section below) were cloned upstream of this minimal promoter fragment. In addition, a shorter fragment of the minimal promoter of 193 bp (-114 to +59 bp) was cloned into pGL3-Basic along with shorter segments of putative regulatory regions. A20 B lymphoma cells were co-transfected with 2 µg of the firefly luciferase plasmids and 0.25 µg of pRL-Renilla luciferase plasmid using Nucleofection (Lonza) and analyzed at 24 hours after transfection. Firefly luciferase activity was normalized to Renilla luciferase activity.

## Results

### The 3-D chromatin organization of the mouse Ets1 gene in B cells

To define topologically-associated domains (TADs) around the mouse *Ets1* gene, we examined Hi-C data from a previously-published study of resting mouse B cells [49]. This analysis showed that the mouse *Ets1* gene is contained within a CTCF-flanked TAD of ~900 kb that stretches from -445 kb to +452 kb from the major transcriptional start site (TSS) of *Ets1* (boundaries of the TAD are shaded in aqua in Fig 1A) [34]. In addition to the *Ets1* gene, this TAD contains fellow Ets family member *Fli1*, as well as two potassium channel-encoding genes, *Kcnj1* and *Kcnj5*. A closer look at the contact map reveals a CTCF-flanked sub-TAD stretching from -107 kb to +244 kb (indicated by the large aqua triangle in Fig 1B). This region contains only the *Ets1* gene itself, with no other known genes. Within the *Ets1* sub-TAD, there is a local hub of more intense interactions that stretches from -107 kb to the end of the Ets1 gene (approximately +62 kb from the TSS) (denoted by the smaller aqua triangle in Fig 1B). It is likely that regulatory elements crucial for B cell-specific expression of mouse *Ets1* are present within the strong interaction hub. We suspect that the CTCF binding site located at -107 kb likely functions to insulate the *Ets1* promoter from regulatory elements of the *Fli1* gene, which is oriented in a head-to-head manner with Ets1 and separated by 154 kb (Fig 1B). Indeed, Hi-C detects little interaction between the *Fli1* promoter and sequences in the *Ets1* sub-TAD.

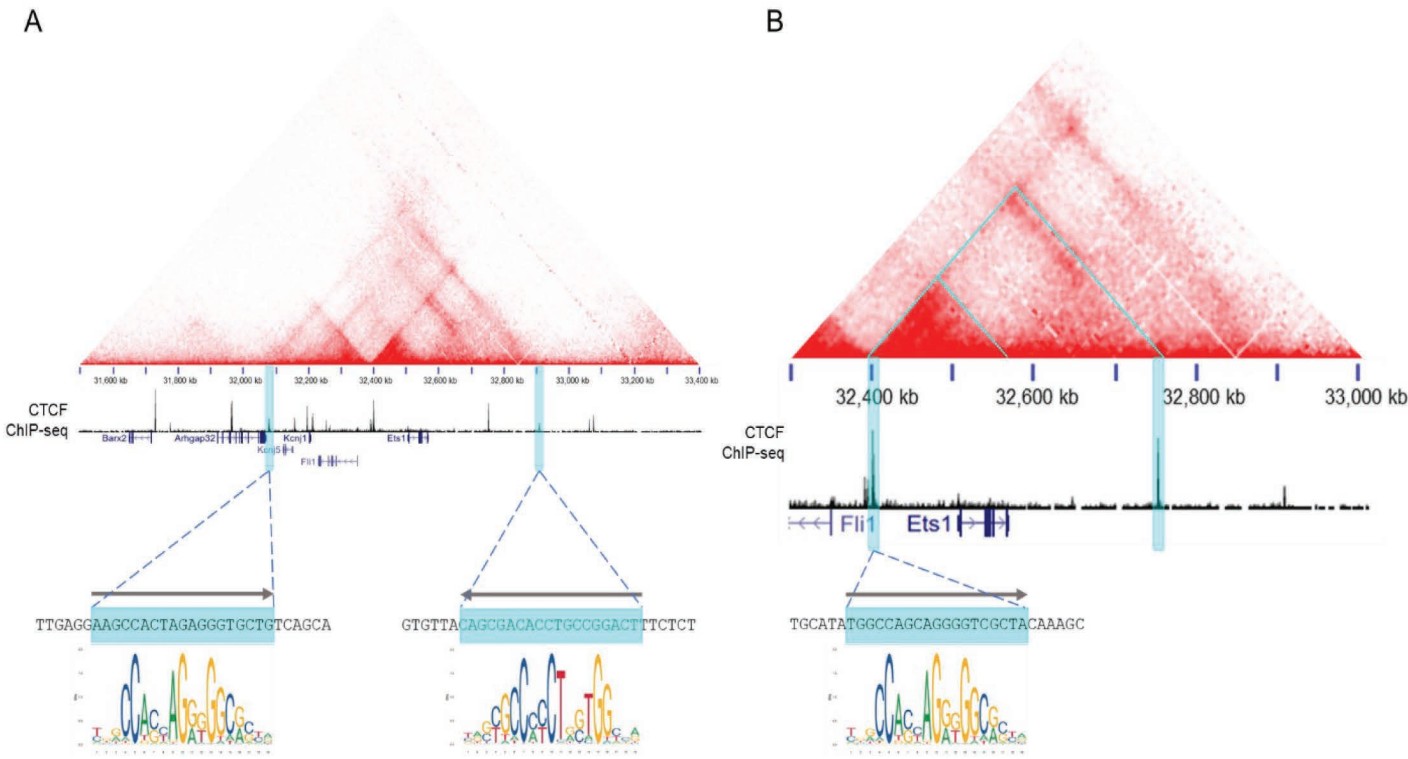

**Fig 1. Identification of the Ets1 TAD and sub-TAD.** Chromatin interaction maps of mouse B cell Hi-C data showing the TAD (A) and sub-TAD (B) that include the *Ets1* gene. The aqua triangles in (B) denote the sub-TAD within the TAD and the interaction hotspot discussed in the text. Below the interaction maps is CTCF ChIP-seq data from splenocytes. The aqua highlights denote the TAD and sub-TAD boundaries that overlap with strong CTCF enrichment. FIMO was used to identify CTCF motifs at the peaks of CTCF binding and are depicted below the CTCF ChIP-seq profiles. The CTCF binding site located at the down-stream boundary of the sub-TAD lacks a consensus CTCF motif and hence no sequence is shown at the bottom for this site.

## A region extending from -113 kb to +105 kb from the mouse *Ets1* TSS -directs expression in B cells

Prior transgenic mouse studies have shown that sequences located from -5.3 kb to +9 kb from the TSS of *Ets1* were not sufficient to drive expression of a transgene in B cells or other lymphoid lineages [50]. To identify sequences that drive B cell-specific expression of *Ets1*, we generated BAC transgenic mice harboring a 219 kb piece of mouse DNA including sequences from -113 kb upstream of the *Ets1* gene (stretching just beyond the CTCF-bound boundary) to +105 kb downstream of the transcriptional start site. This BAC construct contains the entire local hub of robust interactions around the *Ets1* gene (Fig 2A). The first exon of the *Ets1*

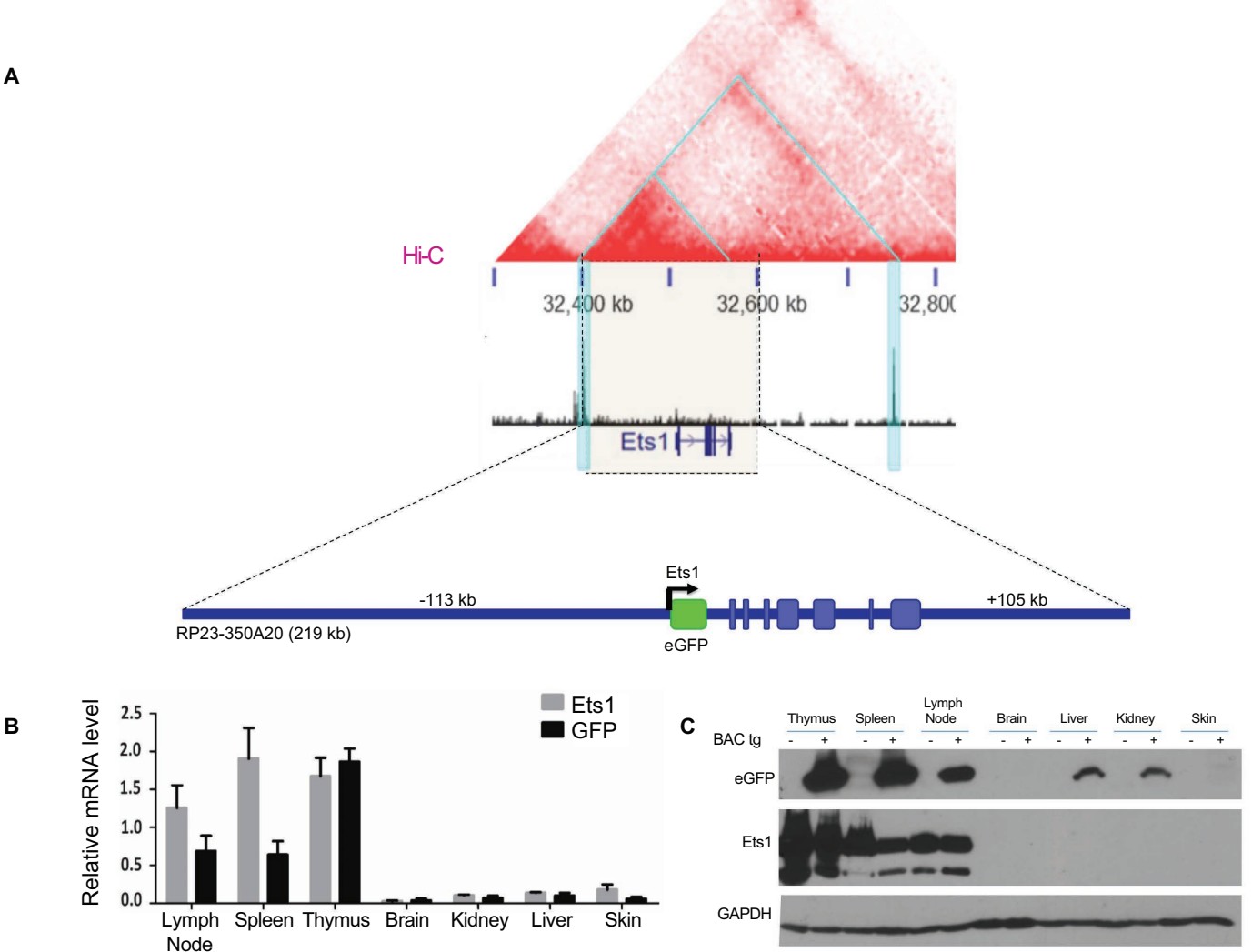

**Fig 2. Design of a BAC-derived reporter construct to identify sequences required for lymphoid Ets1 expression.** (A) Hi-C data from mouse B cells shows that the RP23-350A20 BAC (whose extent is denoted by the light beige block flanked by dotted lines) contains the entire hotspot of most frequent interactions (smaller blue triangle in the top part of the figure). The first exon of Ets1 was replaced in the BAC by eGFP cDNA via recombineering. eGFP reporter expression is under the control of the putative response elements contained with the BAC. (B) qPCR and (C) Western blot to show expression of Ets1 and GFP in various tissues of BACtg mice and non-transgenic mice.

gene was replaced with an eGFP cDNA, followed by a stop codon and an SV40 polyA signal, allowing eGFP expression from the *Ets1* promoter under the direction of putative regulatory elements located within the BAC (Fig 2A). Analysis of both mRNA and protein showed high eGFP expression in lymphoid tissues, including the thymus, spleen, and lymph nodes, where B and T lymphocytes are abundant and where Ets1 is normally expressed at high levels (Fig 2B,2C). Non-lymphoid tissues such as the brain and kidneys showed much lower expression as expected.

To compare the potential overlap in the Ets1 and eGFP expression in these BACtg mice at a cellular level, we performed intracellular flow cytometry using an antibody specific for Ets1. B220$^+$CD19$^+$ B cells from both wild type mice and BACtg mice showed high levels of intracellular Ets1 expression compared to isotype control staining (Fig 3A, bottom). Mature B cell subsets within this B220$^+$CD19$^+$ population, including CD23$^{hi}$CD21$^{lo}$ follicular B cells, CD23$^{lo}$CD21$^{hi}$ marginal zone B cells, and Fas$^+$PNA$^+$ germinal center B cells, all showed strong positive staining for both Ets1 and eGFP (Fig 3B). Ets1 is down-regulated upon B cell differentiation to plasma cells [12,15,51]. In keeping with this, eGFP levels were lower in B220$^{lo}$CD138$^+$CD98$^+$ plasma cells than in B220+CD19+ B cells of BACtg mice (Fig 3B). However, in contrast to the B cells, eGFP expression did not fully reflect Ets1 expression in the T cell populations of BACtg mice. While both CD3$^+$CD4$^+$ and CD3$^+$CD8$^+$ T cells had uniformly high staining for intracellular Ets1, GFP expression showed a broad range in these populations (Fig 3B), indicating variegation of BACtg expression.

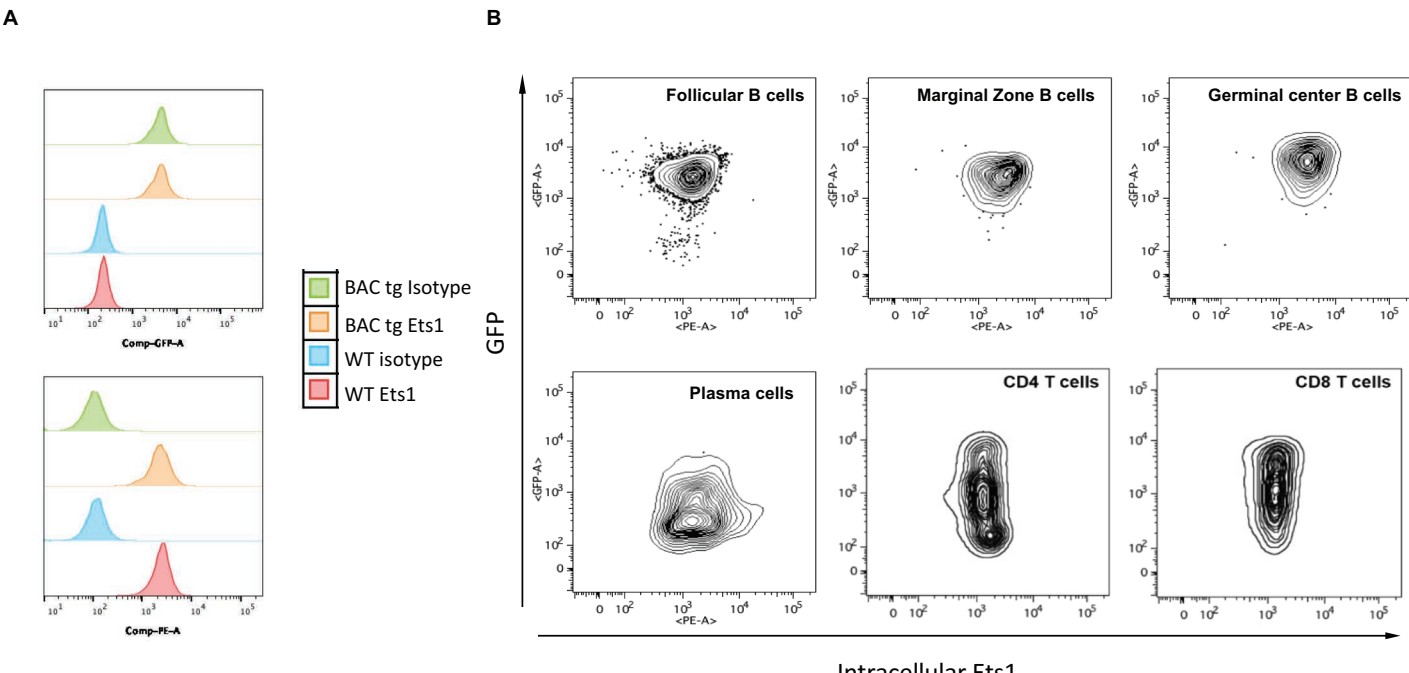

**Fig 3. Expression of the BACtg eGFP in lymphoid cells.** (A) Flow cytometry analysis of intracellular Ets1 and eGFP in gated B220+CD19+ B cells from spleen of wild-type non-transgenic (WT) and BAC transgenic (BACtg) mice. The top figure shows GFP staining while the bottom figure shows intracellular Ets1 staining. (B) Flow cytometry plots of intracellular Ets1 versus eGFP staining in splenic B cell subsets: follicular B cells (B220+CD19+CD23$^{hi}$CD21$^{lo}$), marginal zone B cells (B220+CD19+CD23$^{lo}$CD21$^{hi}$), germinal center B cells (B220+CD19+Fas+PNA+), memory B cells (B220+CD19+CD80+PDL2+), plasma cells (B220$^{lo}$CD138+CD98+), CD4 T cells (CD3+CD4+) and CD8 T cells (CD3+CD8+).

BCR stimulation induces the downregulation of Ets1 in B cells, mainly through a decrease in *Ets1* gene transcription [16,17]. To determine if the response elements required for this phenomenon are within the 219 kb of the BAC, B cells from the BACtg animals were isolated and stimulated *ex vivo*, and expression of Ets1 and eGFP were assessed. As shown in S1 Fig, stimulation by the BCR resulted in a decrease in Ets1 pre-mRNA and mRNA in B cells. Because the pre-mRNA is a short-lived product in the cells, its measurement serves as a surrogate for the transcriptional level of the *Ets1* gene. In the BACtg B cells, eGFP mRNA was similarly downregulated in response to BCR stimulation, although not quite as dramatically. We surmise that the attenuated downregulation of eGFP compared to Ets1 is due to a difference in mRNA stability of the two transcripts [16,52]. Overall, eGFP expression in the mature B cell populations of these BACtg mice recapitulates that of endogenous Ets1 under both basal conditions and upon BCR stimulation.

## Multiple putative regulatory elements have the potential to regulate Ets1

To identify candidate regulatory elements that might control *Ets1* expression in mouse B cells, we examined epigenetic characteristics of regions included in the BAC transgene in more detail. We identified regions that had open chromatin, as defined by peaks of ATAC-seq accessibility in mouse B cells (highlighted in aqua in Fig 4A). H3K4me3 histone methylation, a mark of active promoters, was enriched at the TSS. H3K4me1 histone methylation and H3K27ac histone acetylation, both marks of active regulatory regions [53], were enriched at numerous DNA segments upstream and within the *Ets1* gene. On the other hand, regions downstream of the *Ets1* gene did not harbor any regions that showed enrichment for H3K27Ac or H3K4me1 in B cells. These observations, combined with the Hi-C data shown in Fig 1 and the BAC transgenic data shown in Figs 2, 3, suggest that in mouse B cells *Ets1* is likely regulated by DNA sequences upstream and within the gene itself, rather than sequences located downstream of the gene.

The BAC transgenic mice have variegated expression of eGFP in T cell populations, suggesting that the BAC lacks all the elements required for reproducible expression of Ets1 in T cells. In contrast to B cells which lack putative regulatory elements downstream of the *Ets1* gene, ATAC-seq data from mouse T cell populations shows that T cells have accessible sites downstream of the *Ets1* gene (highlighted in aqua in S2 Fig). These sites are located beyond the downstream regions included in the BAC (highlighted in yellow in S2 Fig) and could represent T cell-specific enhancers required for optimal and sustained expression. Since some of the T cells in BACtg mice express high levels of GFP, the 219 kb region of the BAC must encompass regulatory elements that can induce expression in T cells. However, these elements appear to be insufficient to prevent spreading of heterochromatin into the *Ets1* locus in T cells thus resulting in variegated expression of the transgene. Recently a region located ~ 250 kb downstream of *Ets1* was shown to function as a multi-enhancer for mouse Th1 cells [31].

While peaks of H3K27Ac and H3K4me1 histone marks were discrete in regions 5' to the *Ets1* gene, there was broad enrichment of these marks across the body of the mouse *Ets1* gene (Fig 4A). To better define potentially important regulatory regions within the *Ets1* gene itself, we examined histone marks in a human B cell line GM12878 that expresses Ets1. As shown in Fig 4B, GM12878 cells have a more discrete pattern of enrichment of H3K27Ac at seven specific regions within and around the human *Ets1* gene. The sever regions were designated Sites 1-7 and contained regions of DNA sequence conservation between mouse and human, supporting the idea that they may be functionally important (S3 Fig). These regions of H3K27Ac enrichment are characterized by the presence of DNAse I hypersensitivity (Fig 4B). Site 4 corresponds

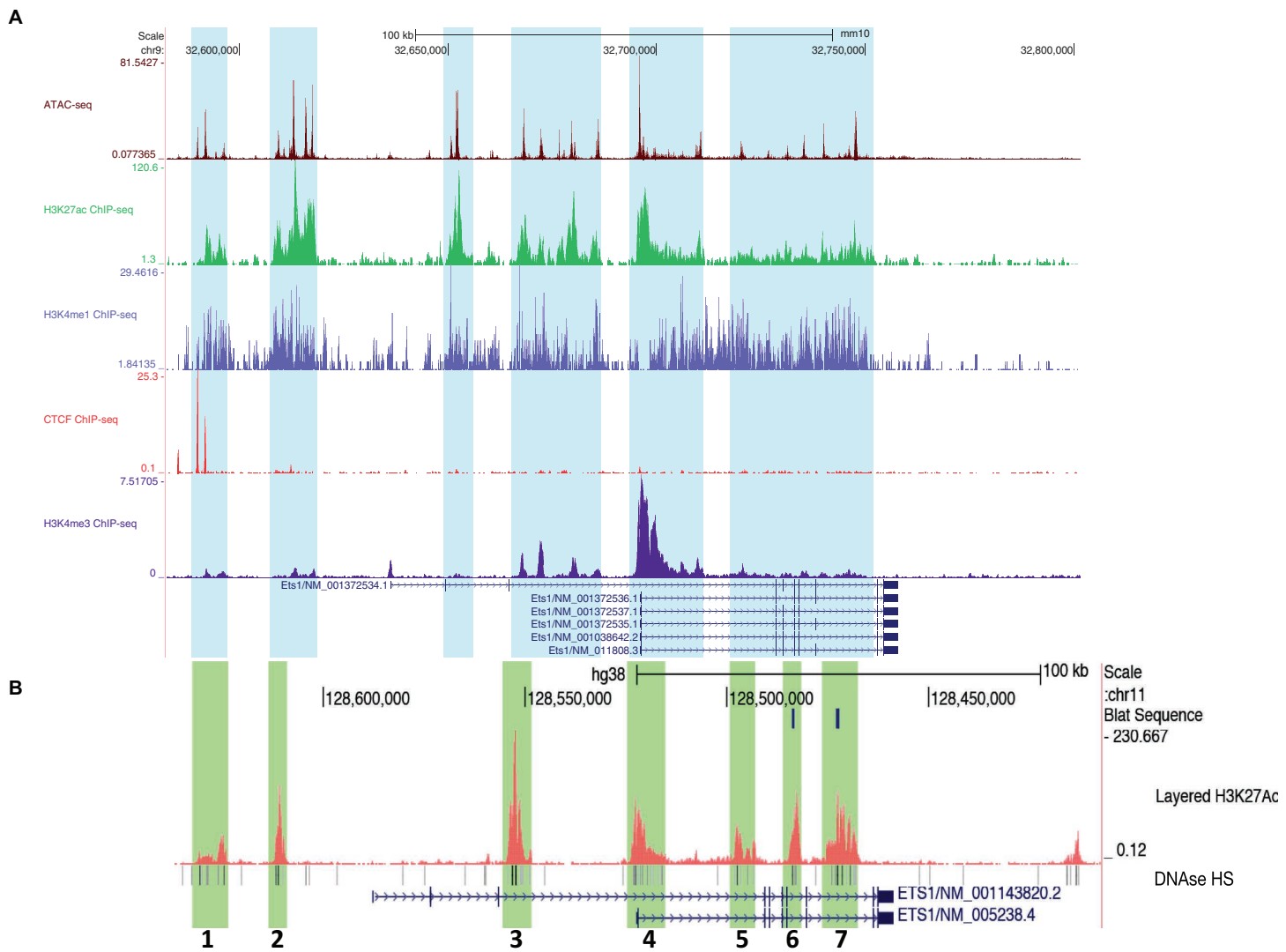

**Fig 4. The Ets1 sub-TAD region contains several putative response elements.** (A) Genome Browser image depicting B cell epigenetic profiles within the Ets1 sub-TAD. Datasets shown include mouse B cell ATAC-seq (accessible chromatin) and ChIP-seq for H3K27ac (active response elements), H3K4me1 (active or poised response elements), CTCF (denoting the upstream sub-TAD boundary), and H3K4me3 (active promoter). Highlighted in light blue are sequences with ATAC-seq peaks and enrichment of H3K27ac and H3K4me1. Several transcript variants of the mouse Ets1 gene are shown, but the major isoform expressed in B cells is Ets1/ NM_011808.3 (B) The human Ets1 gene locus in regions homologous to the mouse BACtg. DNAse I hypersensitivity is shown below. Also shown is H3K27ac ChIP-seq (active response elements) from GM12878 B lymphoma cells, which identifies seven discrete regions of enrichment (green highlights labeled 1-7). Two transcript variants are shown, but the major isoform expressed in B cells is Ets1/NM_005238.4.

to the *Ets1* proximal promoter and beginning of intron 1, sequences previously tested in transgenic mice and shown to not mediate lymphoid-specific expression [50]. Thus, this region was not further analyzed. Site 1 encompasses the CTCF binding peaks located at -104 kb upstream of the mouse *Ets1* promoter. Examining the location of SNPs that have been associated with lupus, we found they cluster in two regions, upstream near Site 1 and within the *Ets1* gene near Site 7 and sequences downstream of it (S4 Fig). These sites are distant from the previously-described T cell-specific super-enhancer region of *Ets1*, which was reported to be enriched for SNPs associated with allergy [31]. In summary, there are multiple potential regulatory elements found within and upstream of the mouse *Ets1* gene that may be crucial for B cell-specific expression.

## Chromatin patterns in resting B cells versus plasma cells

Analysis of previously-published RNA-seq data from sorted purified mouse bone marrow progenitors [37] shows that Ets1 is expressed at low levels in hematopoietic stem cells (HSC), multi-potent progenitors (MPP) and common lymphoid progenitors (CLP), but is elevated once cells commit to the B cell lineage and again reduced in fully-differentiated plasma cells (PC) (Fig 5A). Open chromatin patterns assessed by ATAC-seq in the same sorted populations showed that there were differences in the regions of open chromatin between resting follicular (FO) B cells, where Ets1 levels are high, and PC, where Ets1 levels are low (Fig 5B). The ATAC-seq peaks located near the sub-TAD boundary where CTCF binds (-100 to -106 kb region) and the ATAC-seq peaks located in the proximal promoter of *Ets1* were equally present in both FO B cells and PC. When examining the other ATAC-seq peaks, we found that peaks located at approximately -87 kb, -44 kb, -16 kb, -10 kb, +39 kb and +44 kb were strong in B cells, but reduced in PC (Fig 5B, blue highlighting). The differences in chromatin

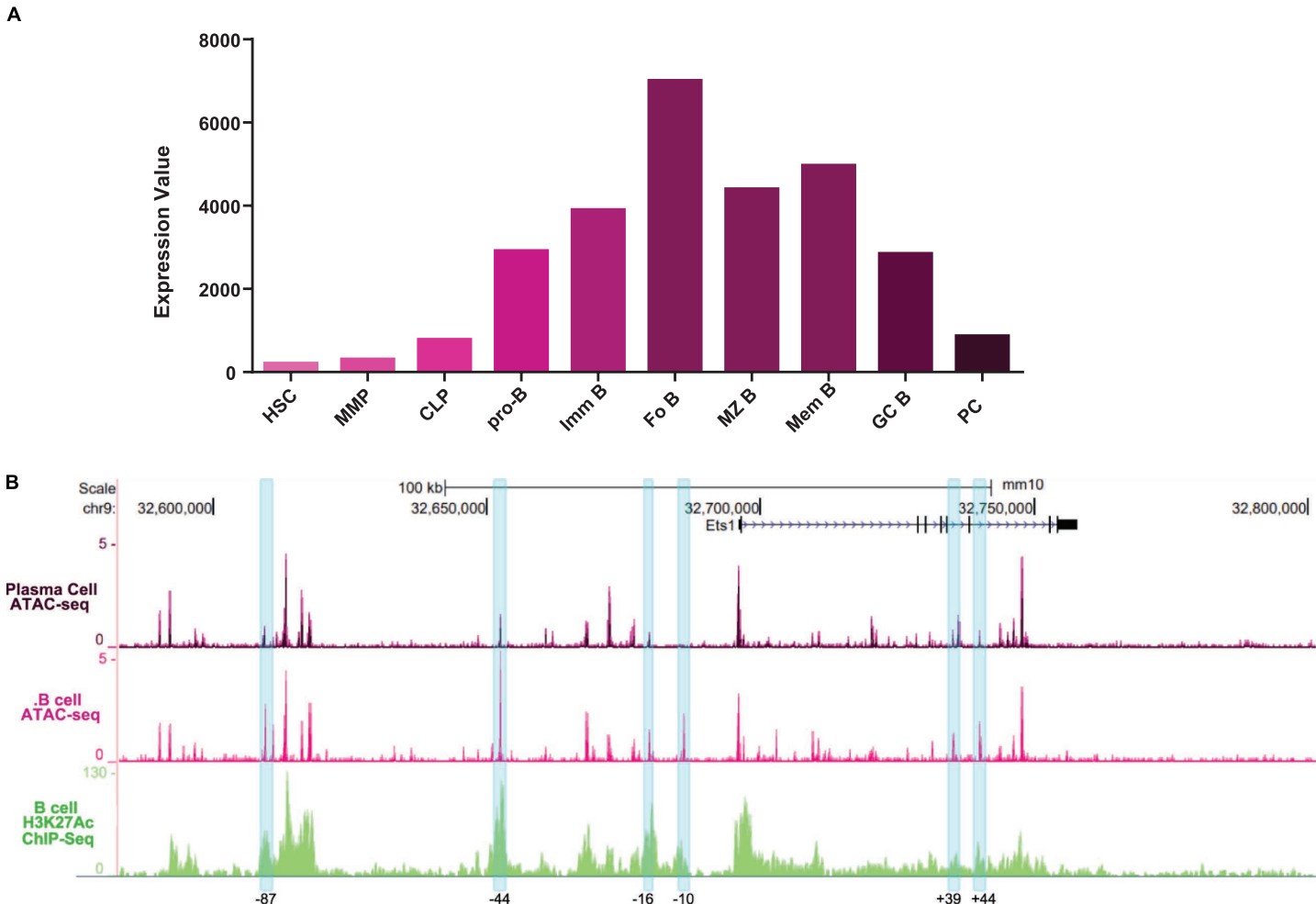

**Fig 5. Differences in Ets1 expression in B cell subsets correlate with changes in chromatin accessibility.** (A) Relative expression levels of Ets1 in sort-purified cells at various stages in the differentiation from hematopoietic stem cells to mature B cells and to plasma cells based on RNA-seq data from the ImmGen immunocyte RNA-seq project (GSE109125). (B) ATAC-seq profiles for follicular (FO B) versus plasma cells (PC), based on ImmGen ATAC-seq data (GSE100738), within the sequences included in the RP23-350A20 BAC. Blue highlighted regions mark six areas with differential ATAC-seq peaks in FO B cells versus PC. Also shown are H3K27Ac peaks from bulk CD3-, B220 +, CD19 + splenic B cells.

accessibility at these regions may indicate the presence of regulatory elements bound by activating transcription factors promoting Ets1 expression in B cells.

## Putative regulatory elements within and upstream of the *Ets1* gene change in accessibility upon BCR stimulation

As mouse *Ets1* gene transcription is decreased in response to BCR stimulation and this effect is mediated by response elements within the BAC, we explored changes in chromatin accessibility in B cells in response to BCR stimulation. The downregulation of *Ets1* gene transcription occurs early after activation, within 2 hours of BCR engagement [16]. We therefore generated ATAC-seq datasets for resting mouse B cells and B cells stimulated for 2 hours with a BCR crosslinking antibody. Regions that experienced a two-fold or greater change in accessibility between resting and stimulated B cells were deemed as differentially accessible regions (DARs) (Fig 6A). Upon stimulation, changes in chromatin accessibility were induced genome-wide, with 2259 DARs increasing and 598 DARs decreasing in accessibility. The disproportionate amount of DARs increasing in accessibility versus those decreasing is in line with previous studies of immune cell *ex vivo* stimulation [54,55]. Genes within 10 kb of DARs that gained or lost accessibility were found to be functionally associated with immune responses and signal transduction (Fig 6B). Motif enrichment analysis revealed that DARs that gained accessibility frequently contained binding sites for AP-1 and NF-κB proteins, which are downstream effectors of pathways activated by BCR signaling, while those that lost accessibility were most highly enriched for the Ets1 motif (Fig 6C). Thus, Ets1 or other members of the Ets gene

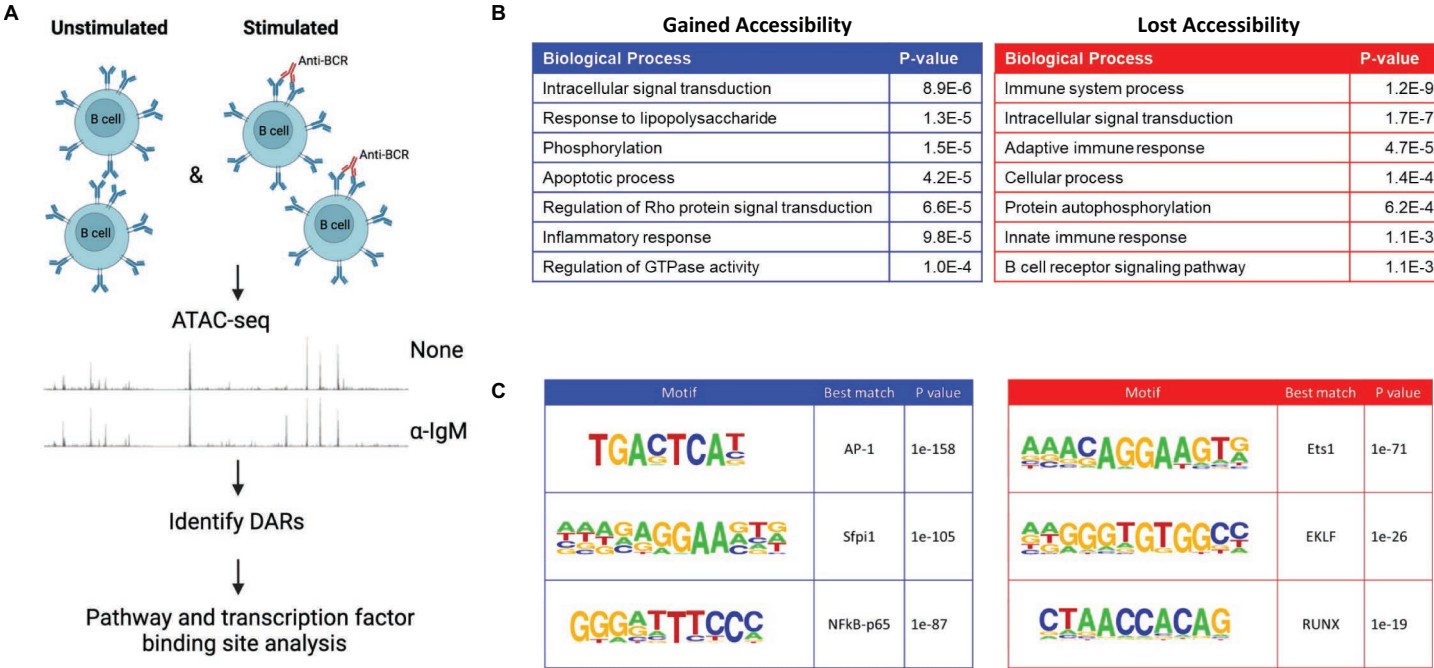

**Fig 6. Identification of differentially-accessible regions (DARs) in response to BCR stimulation.** (A) Schematic of the experimental layout. Total splenic B cells from wild-type C57BL/6 mice were isolated and either left unstimulated or stimulated for 2 hours with BCR crosslinking antibody (anti-IgM). ATAC-seq was performed and DARs were identified. (B) Top seven biological processes identified by analyzing genome-wide DARs that either gained or lost accessibility upon BCR stimulation. Gene ontology analysis was performed on genes within 10 kb of DARs using the DAVID software. (C) Transcription factor binding motifs enriched in DARs that gained (left) or lost (right) accessibility with BCR stimulation. Genome-wide DAR sequences were analyzed by HOMER for *de novo* transcription factor motif discovery. Shown are the top 3 de novo motifs enriched in each set of DARs and the best matching known consensus motif.

family may show decreased expression or activity in stimulated B cells. The loss of Ets1 binding in BCR-stimulated B cells is consistent with the decreased transcription of the *Ets1* gene upon BCR stimulation. However, it is also likely that BCR-induced phosphorylation of the Ets1 protein, which has been shown to decrease its DNA binding activity and in T cells leads to its nuclear export [56,57], contributes to loss of Ets1 binding in stimulated B cells.

Four ATAC-Seq peaks in the region contained within the BACtg were identified as DARs, with 2 gaining and 2 losing accessibility after 2 hours of BCR stimulation (Fig 7A, DARs highlighted in blue and red, respectively). These sites were located at -85 kb (included within Site 2 defined in Fig 4B), at +36 and +39 kb (included within Site 6) and at +49 kb (included within Site 7) from the TSS. The changes in DNA accessibility could reflect stimulation-induced alterations in transcription factor binding. A decrease in accessibility upon stimulation could reflect the loss of binding of a transcriptional activator, while an increase in accessibility may be caused by increased binding of a transcriptional repressor. Both scenarios could lead to downregulation of *Ets1* gene transcription.

DNA sequences in the +39 kb and +49 kb DARs showed conservation with corresponding sequences in human *Ets1* gene (S3 Fig). On the other hand, the other two mouse DAR sequences at -85 kb and +36 kb didn't show strong DNA sequence conservation, although these DARs were located within Sites 2 and 6 that did have regions of conserved sequence. To identify transcription factors that may bind to the four DAR sites, we used the FIMO program to scan the sequences for DNA-binding motifs. The five strongest motif matches for each site are mapped in Fig 7B. Transcription factors sites found within the DARs that lose accessibility after BCR stimulation include those for NF-κB proteins, SpiC, Smad proteins and Irf5, among others. Both sites that gained accessibility contained motifs for AP-1 family members and for Srebf2, a sterol RE-binding protein (Fig 7B). ChIP-seq data from the CH12 mouse B cell lymphoma line indicates that both c-Jun and JunD can bind at the +49 kb DAR site that gains accessibility (Fig 7C) [36]. With the exception of Hoxd3 and Hnf4g, all of the transcription factors shown in Fig 7B are expressed in B cells [37] and could potentially be involved in regulating *Ets1* transcription. These changes in chromatin accessibility that occur in response to BCR stimulation could highlight the response elements important for regulation of Ets1 expression and implicate AP-1, in particular, in its downregulation.

### A selection of regulatory elements tested in transient transfections are insufficient to activate the *Ets1* promoter

To determine if the potential regulatory elements identified above govern Ets1 expression, we next generated a series of luciferase reporter constructs with putative regulatory elements cloned upstream of the *Ets1* minimal promoter (Fig 8). We generated two sets of constructs. In the first set, a 469 bp fragment of the *Ets1* minimal promoter (451 bp to +18 bp from the TSS) was cloned along with putative regulatory element fragments containing regions of Sites 1-3 and 5-6 (shown as blue boxes in Fig 8). The second set of constructs was more focused and included a smaller fragment of the *Ets1* promoter (−114 bp to +60 bp from the TSS) along with shorter segments of the putative regulatory elements (shown in green boxes in Fig 8). For the second set of constructs, we chose elements that were located in regions of H3K27ac enrichment and that were more accessible in either resting B cells or plasma cells (as highlighted in Fig 5B), assuming that differences in chromatin accessibility would reflect the high and low expression of Ets1 in these cell types.

The reporter constructs were transfected into the A20 mouse B lymphoma cell line, which expresses Ets1 and downregulates it transcriptionally in response to BCR stimulation similar to primary B cells [16,17]. Out of the putative regulatory elements tested, ranging from

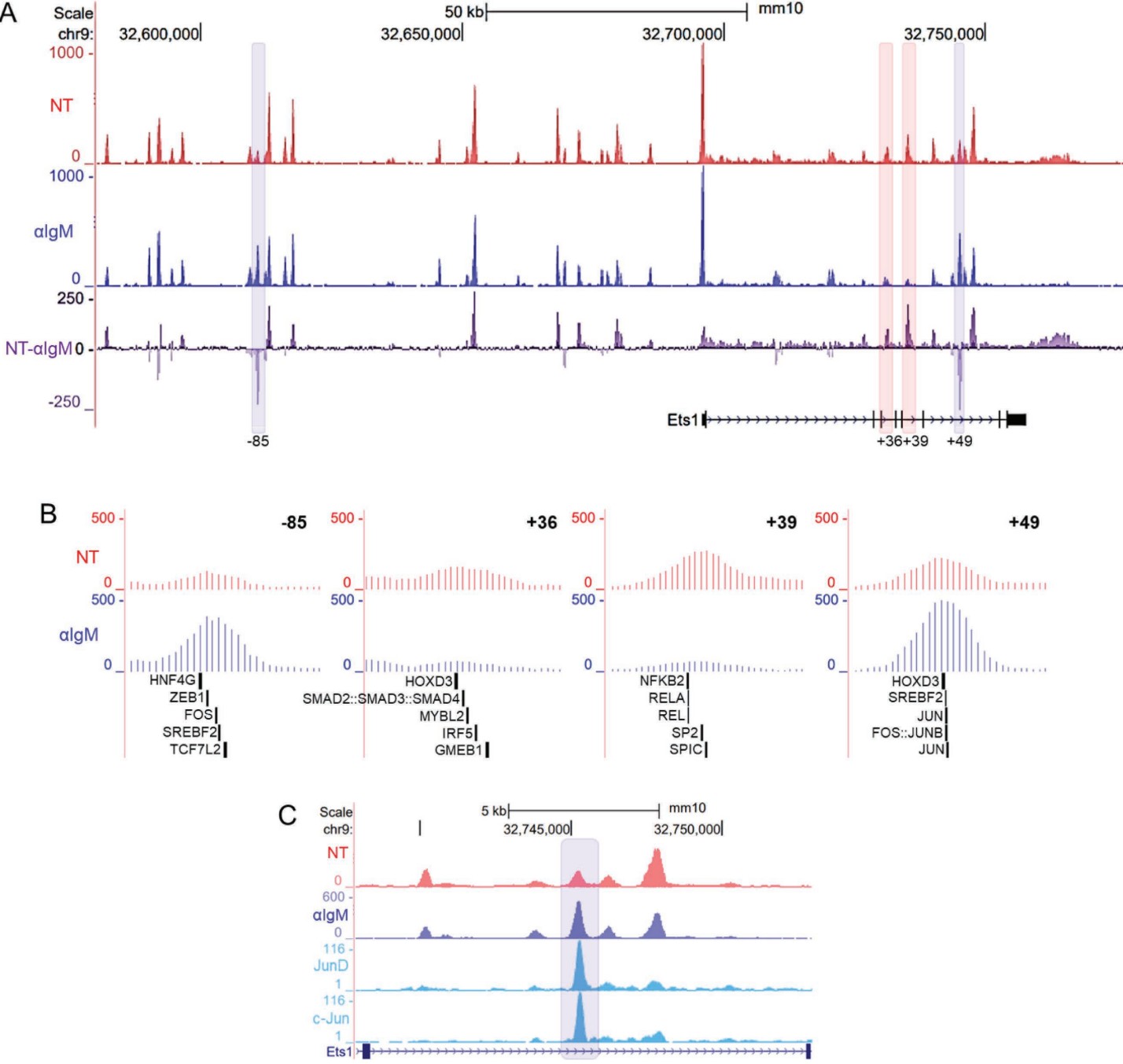

**Fig 7. DARs in the *Ets1* gene locus contain motifs for B cell-relevant transcription factors.** (A) Shown are the ATAC-seq profiles in the Ets1 locus (within the region of the BAC transgene) in unstimulated (NT) and stimulated (αIgM) B cells. The third track shows the difference between the two datasets, with the positive peaks indicating sites that are more accessible in unstimulated and negative peaks indicating sites that are more accessible in stimulated B cells. DARs showing at least 2-fold changes in accessibility are highlighted blue for those that gained accessibility and highlighted in red for those that lost accessibility. (B) Close-up views of the DARs and transcription factor motifs they contain. FIMO was used to scan for transcription factor motifs within the four DARs. Depicted are the five highest scoring motifs. (C) ATAC-seq profiles compared with AP-1 subunits (c-Jun and JunD) ChIP-seq data from CH12 cells. The blue highlight denotes the +49 kb DAR.

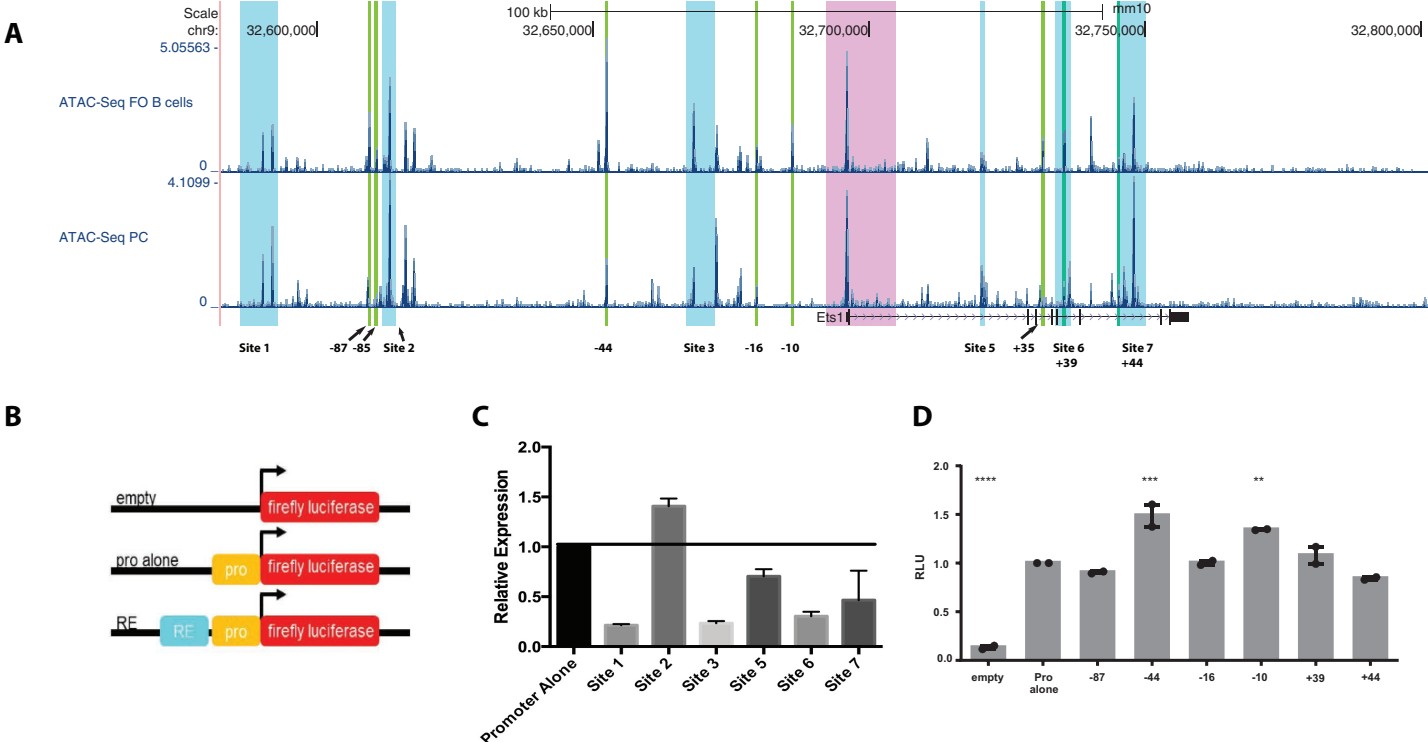

**Fig 8. Individual putative response elements do not mediate strong activation or repression in transient transfection assays.** (A) ATAC-seq profiles of follicular (FO) B cells and plasma cells (PC) in the *Ets1* locus (the region contained within the BAC transgene). Highlighted in light blue are larger fragments equivalent to Sites 1-3 and 5-7 that were incorporated into luciferase constructs. Highlighted in bright green are small regions containing differentially-accessible regions comparing follicular B cells to plasma cells (as shown in Fig 5B). Highlighted in pink is the region surrounding the proximal promoter and part of first intron that were previously tested in transgenic mice and shown to be insufficient for mediating lymphocyte-specific expression. (B) General schematic of the design of the luciferase constructs. (C-D) A20 cells were transfected with firefly luciferase plasmids containing the Ets1 promoter alone or with the indicated response elements, or with empty vector, along with eF1α promoter Renilla luciferase internal control plasmid. Luciferase activity was measured 24 hours after transfection. Firefly luciferase activity was normalized to Renilla luciferase activity, then values were set relative to the plasmid with the Ets1 promoter alone. Significance was determined by one-way ANOVA. N = 2-5 replicates for each transfection.

-110 kb upstream to +49 kb downstream of the *Ets1* TSS, none showed a strong ability to activate expression (Fig 8C,8D), although some sequences appeared able to repress transcription. The strongest activity was found with sequences in Site 2 that were able to stimulate transcription modestly at approximately 1.5-fold over the promoter alone (Fig 8C). Given that none of these sequences alone was able to direct high level expression of a reporter gene in B cells, these data suggest that Ets1 may be regulated by a combinatorial network of response elements that cooperate to stimulate gene expression in B cells.

## Discussion

In mice, loss of Ets1 leads to autoimmune disease with inappropriate activation of B cells, including autoreactive B cells, and their differentiation into plasma cells that secrete antibodies [8,15,16]. Numerous autoimmune disease-associated SNPs map near the human *Ets1* gene [1] and decreased Ets1 expression has been found in immune cells from autoimmune disease patients [2–7,58,59]. Given that aberrant expression of the *Ets1* gene is found in autoimmune diseases, it is important to understand the regulatory mechanisms controlling its expression pattern.

Previous studies to examine cell-type specific regulation of the *Ets1* gene focused on sequences ranging from -5.3 kb to +9 kb with respect to the major transcriptional start site of

the gene. However, these sequences fail to drive lymphocyte expression of a reporter gene in transgenic mice [50]. To determine regulatory elements for *Ets1* in B cells, we first identified an 835 kb topologically-associated domain (TAD) flanked by CTCF binding sites that contains the mouse *Ets1* and other genes. This large TAD region includes a smaller 289 kb CTCF-flanked sub-TAD containing only the *Ets1* gene. Within the sub-TAD, Hi-C analysis shows that B cells have an interaction hotspot extending from approximately -110 kb to the end of the final exon of *Ets1*. The TAD structure of the mouse *Ets1* gene is conserved in CD4 + T cells, as shown by Chandra et al [31].

We generated a BAC-transgenic reporter system carrying sequences from -113 kb to +105 kb from the *Ets1* TSS, which contains the Hi-C interaction hotspot of *Ets1*. Sequences in the BAC transgene recapitulated well the expression of Ets1 in B cells, indicating that all regulatory elements needed for B cell expression are present within the interaction hotspot. B cell activation by BCR crosslinking or incubation with TLR ligands leads to downregulation of Ets1 and this is mediated mainly at the level of gene transcription and does not require new protein synthesis [16,17]. The sequences contained in the BACtg are sufficient to mediate downregulation of the GFP reporter in response to BCR stimulation. In contrast to the pattern in B cells, the expression of eGFP in T cells from BACtg mice was variegated, suggesting the existence of further T cell-specific enhancers outside the BAC segment. Variegation of gene expression is typically due to stochastic spread of heterochromatin in a subset of cells. Thus, there may be sequences located downstream of the 3' end of the BAC that are required in T cells, but not B cells, to prevent heterochromatin spread. A previously-described super-enhancer segment located ~ 250 kb downstream of *Ets1* was shown to be essential for maximal Ets1 expression in double positive thymocytes and in Th1 cells and for proper Th1 cell differentiation [31]. This super-enhancer region is not present in the BACtg and its absence may lead to variegated expression of the GFP marker in T cells. However, it is important to note that there is also a region of accessible chromatin in T cells located at approximately +120 kb from the *Ets1* TSS that is missing from the BACtg (see S2 Fig). This sequence could also play a role in regulation of *Ets1* expression in T cells.

To identify potential regulatory elements of Ets1, we examined epigenetic datasets from B cells and defined seven broad regions that have properties of regulatory sequences (Sites 1-7). All seven of these sites have regions with high DNA sequence homology between the mouse and human genomes. However, these large regions of sequence showed only weak activity in a luciferase reporter assay in A20 B lymphoma cells. To refine our analysis, we compared open chromatin regions in follicular B cells, which express high levels of Ets1, to open chromatin regions in plasma cells, which have very low levels of Ets1. This analysis identified 6 regions with differential accessibility, two of which overlapped with sequences in Sites 1-7. We tested these six shorter segments in luciferase assays, but they also failed to mediate strong activation of the promoter. Therefore, we did not identify any regions that were able to strongly activate or repress expression of a reporter gene in transient transfection assays. This result suggests the possibility that transcription of the Ets1 gene is more complicated and relies on the combinatorial action of multiple regulatory regions that must function cooperatively.

Since sequences in the BACtg were sufficient to mediate down-regulation of *Ets1* gene transcription upon BCR stimulation, we identified differentially-accessible regions (DARs) in the chromatin of stimulated and unstimulated B cells. In the region surrounding the Ets1 locus, four DAR regions were identified and three of these overlapped with sequences in Sites 1-7. The two sites that gained accessibility after stimulation contained motifs for AP-1 family members. AP-1 motifs were also enriched globally in sites that gained accessibility with stimulation, in line with other studies examining stimulation-responsive chromatin and gene expression that have implicated AP-1 in mediating early genome-wide changes [60, 61].

AP-1 is activated downstream of MAPK pathways, particularly JNK and ERK, and we have previously demonstrated that JNK activity is required for *Ets1* downregulation [16]. Binding of AP-1 subunits c-Jun and JunD was detected at the +49 DAR in CH12 B lymphoma cells. While AP-1 proteins often serve to activate gene expression, considerable evidence shows that they can also serve as repressors of transcription in appropriate contexts [62–66]. Thus, recruitment of AP-1 proteins to the -85 kb and +49 kb DAR regions might serve to inhibit Ets1 gene transcription after BCR stimulation. Further studies will be needed to fully determine the role of AP-1 family proteins in the regulation of Ets1 expression.

The two regions that lost accessibility upon stimulation could represent regulatory elements bound by transcription factors responsible for maintaining Ets1 expression in resting B cells. These two regions contained potential binding sites for a variety of transcription factors including Smad proteins, NFκB family members, and Ets family member SpiC. SpiC is a member of the Ets family that has been shown to be a negative regulator of B cell function and could potentially play a role in regulation of Ets1 expression [67]. Canonical NFκB proteins are sequestered in the cytoplasm in the resting state and hence seem unlikely to mediate basal expression of Ets1 in resting B cells. On the other hand, it is possible that non-canonical NFκB signaling, such as that triggered by the BAFF receptor or CD40 receptor, contributes to the maintaining Ets1 expression in resting B cells. The Smad2/3/4 complex is activated downstream of TGFβ signaling, which has been shown to suppress B cell responses through upregulation of inhibitory signaling components [68], and mice lacking the receptor TGFβ RII have hyper-responsive B cells [69]. Therefore, the TFGβ signaling pathway is an attractive possibility for regulating basal levels of Ets1 in resting B cells. Future studies on potential roles for Smad proteins and non-canonical NFκB proteins will be important to determine whether they are involved in maintaining Ets1 in resting B cells.

We evaluated the activity of putative enhancer regions using transient transfection assays with a luciferase reporter gene. While such assays can provide valuable insights, they have limitations, including the lack of native chromatin context. Furthermore, our assays did not assess potential cooperativity between putative regulatory regions. Notably, we did not identify any enhancer regions capable of stimulating transcription in the transient transfection assays. Future studies will more effectively examine the roles of individual regulatory elements by employing CRISPR/Cas9-mediated modifications to delete and/or mutate putative enhancer segments within their natural chromosomal context.

The transcription factors responsible for maintaining high levels of Ets1 expression in resting B cells remain unclear. Our previous work showed that Foxo3a is not required for basal Ets1 expression or its downregulation upon activation [17]. Both BCR and TLR ligation reduce Ets1 expression in B cells, and we have demonstrated that IKK2 signaling is essential for this process [17]. However, neither RelA nor cRel is individually necessary for Ets1 downregulation. Ongoing studies aim to identify the specific transcription factors involved in maintaining high Ets1 levels in resting B cells, as well as those responsible for its downregulation in activated B cells.

Overall, in this report we have analyzed the regions surrounding the mouse *Ets1* gene and determined that a BAC transgene carrying sequences from −113 kb to +105 kb from the TSS is sufficient for high level and reproducible expression of a reporter gene in B cells. Unexpectedly, this region is not sufficient for maintaining consistent expression of the reporter gene in T cells and instead resulted in variegated expression in both CD4 and CD8 T cells. Therefore, there are differences in the regulatory sequences required for optimal expression in B and T cells. Further analysis of epigenetic marks and the comparison of open chromatin in resting B cells versus activated B cells as well as in follicular B cells versus plasma cells resulted in the identification of numerous regions with features of regulatory elements that could participate

in controlling Ets1 expression in B cells. However, none of the elements was sufficient by itself to strongly activate or repress transcription in transient transfection assays, suggesting that *Ets1* may be regulated by a network of several regulatory elements located upstream and within its introns. One limitation worth pointing out is the episomal nature and the lack of intact chromatin conformations in plasmid-based reporters that were utilized in this report. Further studies are needed to fully dissect the *Ets1* cis-regulatory network and the role it may play in B cell activation.

## Supporting information

**S1 Fig. Sequences within the RP23-350A20 BAC can mediate the BCR-induced decrease in transcription from the Ets1 promoter.** Splenic B cells were isolated from wild type and BACtg mice, and were incubated with BCR crosslinking antibody for 2 hrs. Ets1 mRNA and pre-mRNA levels as well as GFP mRNA levels were measured using qPCR and were normalized to beta-actin. N = 2 for each condition.
(TIF)

**S2 Fig. ATAC-seq shows that T cells contain regions of open chromatin that lie outside the region of the BACtg.** (B) ATAC-seq profiles for B cells, CD4 T cells, and CD8 T cells within the *Ets1* sub-TAD. The genomic region contained in the BAC transgene is highlighted in yellow. Highlighted in aqua are accessible sites outside of the BAC that may be T cell-specific *Ets1* response elements.
(TIF)

**S3 Fig. Sequence conservation between mouse and human genes encoding Ets1 in regions with putative response elements.** Genome Browser image depicting the ATAC-seq peaks in follicular (FO) B cells and plasma cells (PC). Orange highlighted regions are regions within Sites 1-7 that show strong sequence homology between mouse and human. Underneath are shown BLAT alignments for these segments. DNA sequence homologies outside Sites 1-7 are not shown. Also shown are the locations of the four DAR regions identified in Fig 7A, with the + 39 and + 49 kb DARs showing significant sequence conservation, while the -85 and + 36 kb DARs lack such conservation.
(TIF)

**S4 Fig. Locations of lupus-associated SNPs in the human Ets1 gene.** Genome Browser image depicting ATAC-seq peaks in human CD19 + B cells (top) and the H3K27Ac peaks from human GM12878 B lymphoma cells (the same peaks used to define Sites 1-7). At the bottom are shown positions of systemic lupus erythematosus (SLE) associated SNPs in the gene. Also shown is the location of the previously-defined *Ets1* super-enhancer region where many allergy-associated SNPs are localized [31].
(TIF)

## Acknowledgments

We thank Alex Glather, Akinsola Oyelakin and Dr. Yungki Park (all at State University of New York at Buffalo) for helpful discussions relevant to this study. We thank Kirsten Smalley for help with animal husbandry.

## Author contributions

**Conceptualization:** Alyssa Kearly, Prontip Saelee, Satrajit Sinha, Anne Satterthwaite, Lee Ann Garrett-Sinha.

**Data curation:** Alyssa Kearly, Prontip Saelee, Jonathan Bard.

**Formal analysis:** Alyssa Kearly, Prontip Saelee, Jonathan Bard, Lee Ann Garrett-Sinha.

**Funding acquisition:** Anne Satterthwaite, Lee Ann Garrett-Sinha.

**Investigation:** Alyssa Kearly, Prontip Saelee, Lee Ann Garrett-Sinha.

**Methodology:** Alyssa Kearly, Prontip Saelee, Jonathan Bard, Satrajit Sinha, Anne Satterthwaite, Lee Ann Garrett-Sinha.

**Project administration:** Anne Satterthwaite, Lee Ann Garrett-Sinha.

**Resources:** Lee Ann Garrett-Sinha.

**Supervision:** Lee Ann Garrett-Sinha.

**Writing – original draft:** Alyssa Kearly.

**Writing – review & editing:** Anne Satterthwaite, Lee Ann Garrett-Sinha.

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
