## [Decision Letter · Decision Letter 0]

27 Sep 2024

PONE-D-24-32251Sequences within and upstream of the mouse Ets1 gene drive high level expression in B cells, but are not sufficient for consistent expression in T cellsPLOS ONE

Dear Dr. Garrett-Sinha,

Thank you for submitting your manuscript to PLOS ONE. First, I have to apologize for the delay in the reviewing process. After very careful consideration, and revision by an expert in the field, we feel that it has merit but does not fully meet PLOS ONE’s publication criteria as it currently stands. Therefore, we invite you to submit a revised version of the manuscript that addresses the points raised below.

We look forward to receiving your revised manuscript.

Kind regards,

Roberto Mantovani

Academic Editor

PLOS ONE

Journal requirements: 1. When submitting your revision, we need you to address these additional requirements. Please ensure that your manuscript meets PLOS ONE's style requirements, including those for file naming. The PLOS ONE style templates can be found at https://journals.plos.org/plosone/s/file?id=wjVg/PLOSOne_formatting_sample_main_body.pdf and https://journals.plos.org/plosone/s/file?id=ba62/PLOSOne_formatting_sample_title_authors_affiliations.pdf 2. We note that the grant information you provided in the ‘Funding Information’ and ‘Financial Disclosure’ sections do not match.  When you resubmit, please ensure that you provide the correct grant numbers for the awards you received for your study in the ‘Funding Information’ section. 3. Thank you for stating the following financial disclosure:  [NIAID R01 AI122720NCI P30CA016056].  Please state what role the funders took in the study.  If the funders had no role, please state: ""The funders had no role in study design, data collection and analysis, decision to publish, or preparation of the manuscript."" If this statement is not correct you must amend it as needed. Please include this amended Role of Funder statement in your cover letter; we will change the online submission form on your behalf. 4. Please expand the acronym “NIAID and NCI” (as indicated in your financial disclosure) so that it states the name of your funders in full.This information should be included in your cover letter; we will change the online submission form on your behalf. 5. Thank you for stating the following in the Acknowledgments Section of your manuscript: [We thank Alex Glather, Akinsola Oyelakin and Dr. Yungki Park (all at State University of New York at Buffalo) for helpful discussions relevant to this study. We thank Kirsten Smalley for help with animal husbandry. This work was funded by NIH grant R01 AI122720.]We note that you have provided funding information that is not currently declared in your Funding Statement. However, funding information should not appear in the Acknowledgments section or other areas of your manuscript. We will only publish funding information present in the Funding Statement section of the online submission form. Please remove any funding-related text from the manuscript and let us know how you would like to update your Funding Statement. Currently, your Funding Statement reads as follows:  [NIAID R01 AI122720NCI P30CA016056].  Please include your amended statements within your cover letter; we will change the online submission form on your behalf. 6. We note that you have included the phrase “data not shown” in your manuscript. Unfortunately, this does not meet our data sharing requirements. PLOS does not permit references to inaccessible data. We require that authors provide all relevant data within the paper, Supporting Information files, or in an acceptable, public repository. Please add a citation to support this phrase or upload the data that corresponds with these findings to a stable repository (such as Figshare or Dryad) and provide and URLs, DOIs, or accession numbers that may be used to access these data. Or, if the data are not a core part of the research being presented in your study, we ask that you remove the phrase that refers to these data. 7. Please include captions for your Supporting Information files at the end of your manuscript, and update any in-text citations to match accordingly. Please see our Supporting Information guidelines for more information: http://journals.plos.org/plosone/s/supporting-information. 

Reviewers' comments:

Reviewer's Responses to Questions

**Comments to the Author**

1. Is the manuscript technically sound, and do the data support the conclusions?

Reviewer #1: Yes

2. Has the statistical analysis been performed appropriately and rigorously? 

Reviewer #1: Yes

3. Have the authors made all data underlying the findings in their manuscript fully available?

Reviewer #1: Yes

4. Is the manuscript presented in an intelligible fashion and written in standard English?

Reviewer #1: Yes

5. Review Comments to the Author

Reviewer #1: In this paper, the authors set out to establish the regulatory regions that control Ets1 expression in B and T cells. Through combining chromatin state mapping and BAC transgenesis, they identify a region responsible for Ets-1 expression in B cells and establish that additional sequences outside this region are required for robust expression in T cells. The latter are not pursued. Functionally, the authors test various potential regulatory regions for enhancer activity but none are obviously active in that context. Furthermore, motif analysis identifies potential TF regulators but no experiments are provided to test their functionality in B cells. Overall, this study therefore contributes to our understanding of Ets1 expression control in immune cells but is rather preliminary in nature, stopping short of providing any detailed mechanistic insights rather than the current inferences.

Generally, the data as presented are okay but there are a few places for improvement:

(1) As indicated above, some functional testing of potential TF regulators would be a step forwards.

(2) As the luciferase reporter assays are generally none informative, a better strategy would be to engineer changes in the regulatory regions either in the context of the BAC or via CRISPR in appropriate cells line models.

(3) In Fig. 4B, adding the human data is okay but the rationale in the text is not really correct. Ie this does not allow “better definition of the regions” but instead provides evidence for evolutionary conservation.

(4) In Fig. 5B, the second track should be labelled as Fo B cells. Also, what population of B cells does the bottom track come from?

(5) In Fig. 6C, the percentage of regions containing each motif should be added.

In the left table, the authors should comment on the second motif that resembles at ETS motif. On the right side, they cannot conclude this is Ets1 rather than another ETS TF, therefore the results text describing this needs rewording.

(6) I was confused by Fig. 8C. It appears here that several regions seem to be repressive in nature (ie less than promoter alone), so not sure why the results text says that none of the regions show strong repressive activity.

(7) Given that GWAS signals suggest the importance of the Ets1 locus, do any of the SNPs map to putative regulatory elements highlighted in this study? If so what is their predicted effect on activity (ideally would also be tested but as a minimum discussed).

6. PLOS authors have the option to publish the peer review history of their article (what does this mean? ). If published, this will include your full peer review and any attached files.

**Do you want your identity to be public for this peer review?** For information about this choice, including consent withdrawal, please see our Privacy Policy .

Reviewer #1: No

---

## [Author Response · Author response to Decision Letter 1]

12 Nov 2024

Response to Editor Comments:

We have reformatted the manuscript to meet the PLoS One style and formatting. We included information about grant funding in the Cover Letter, as requested. Information about grant funding was removed from the Acknowledgements section. We removed reference to data not shown. We included captions for your Supporting Information files at the end of our manuscript, but also embedded them into the Supplemental Figures.

Response to Reviewer Comments:

Response to Reviewer Comments

Reviewer #1: In this paper, the authors set out to establish the regulatory regions that control Ets1 expression in B and T cells. Through combining chromatin state mapping and BAC transgenesis, they identify a region responsible for Ets-1 expression in B cells and establish that additional sequences outside this region are required for robust expression in T cells. The latter are not pursued. Functionally, the authors test various potential regulatory regions for enhancer activity but none are obviously active in that context. Furthermore, motif analysis identifies potential TF regulators but no experiments are provided to test their functionality in B cells. Overall, this study therefore contributes to our understanding of Ets1 expression control in immune cells but is rather preliminary in nature, stopping short of providing any detailed mechanistic insights rather than the current inferences.

Generally, the data as presented are okay but there are a few places for improvement:

(1) As indicated above, some functional testing of potential TF regulators would be a step forwards. (2) As the luciferase reporter assays are generally none informative, a better strategy would be to engineer changes in the regulatory regions either in the context of the BAC or via CRISPR in appropriate cells line models.

We agree with the reviewer that further studies are needed to identify relevant transcription factors and enhancer sequences needed for B cell expression. In this regard, Our lab has shown that signaling via the kinase IKK is required to downregulate Ets1 expression in response to BCR or TLR activation in B cells, although individually neither RelA nor cRel was required (see our published manuscript – PMID: 36445360). We also found that Foxo3a was not required for either the basal expression of Ets1 nor its downregulation in response to BCR ligation. We are continuing to perform follow-up experiments to examine roles for additional transcription factors (Pax5, Irf4, Ebf1, Bach2, PU.1 and Foxo1) that are expressed at high levels in resting B cells and that might be involved in regulating the basal levels of Ets1, but these studies have yet to yield conclusive results. We posit that combinatorial actions of transcription factors are likely to be at play, just as the case with the enhancers that we report in this study

The reviewer brings up a valid point about the limitations of the transient transfection system with luciferase reporter genes. We are currently developing Crispr systems to mutate putative enhancers in a B cell line. However, these studies are at a very early stage and we are planning to include them in a future publication examining Ets1 regulation in more detail.

Despite the fact that the studies reported in this manuscript have not yet identified a specific master transcription factor or enhancer segments crucial for regulating Ets1 expression, we feel that our studies still add valuable information to the knowledge of regulatory mechanisms that govern Ets1 expression. We have found that the Ets1 gene is regulated differently in B cells and T cells, something that was previously unknown. We have also found that sequences within and upstream of Ets1 are sufficient both for faithful expression in B cells, but also for stimulation induced decrease in Ets1 transcription as well as the decrease that occurs in plasma cells. We have also identified potential regulatory regions for Ets1 as marked by ATAC-seq peaks and histone marks. We have identified changes in chromatin accessibility that occur in B cells in response to BCR ligation – this experiment being a genome-wide study offers a valuable resource. While we have focused on describing such changes in the Ets1 locus, our ATAC-seq data have been submitted to the GEO databank and are available for other researchers to examine when assessing how BCR signaling alters chromatin in various gene loci.

Finally, it is also important to note that journal information for PLoS One states that “We evaluate research on the basis of scientific validity, strong methodology, and high ethical standards—not perceived significance. Multidisciplinary and interdisciplinary research, replication studies, negative and null results are all in scope”. We feel that our manuscript currently meets this criterion for publication.

(3) In Fig. 4B, adding the human data is okay but the rationale in the text is not really correct. i.e. this does not allow “better definition of the regions” but instead provides evidence for evolutionary conservation.

We thank the reviewer for their thoughtful insights. We have modified this section of the text to better reflect the rationale.

(4) In Fig. 5B, the second track should be labelled as Fo B cells. Also, what population of B cells does the bottom track come from?

We have relabeled the figure. The bottom track in this figure is from data collected as part of this study (PMID: 25103404). It represents bulk CD3-, B220+, CD19+ B cells isolated from mouse spleen, which has been noted.

(5) In Fig. 6C, the percentage of regions containing each motif should be added.

In the left table, the authors should comment on the second motif that resembles at ETS motif. On the right side, they cannot conclude this is Ets1 rather than another ETS TF, therefore the results text describing this needs rewording.

These motifs were discerned using the motif-finding algorithm HOMER (http://homer.ucsd.edu/homer/introduction/basics.html). HOMER scores motifs by looking for sequences with differential enrichment between two sets of sequences. In our case these two sets are DAR sequences from unstimulated versus stimulated B cells. HOMER calculates a p value for enrichment of known transcription motifs, but does not provide a value for the percentage of sequences containing a particular motif. Given the very small p values of the listed transcription factors, it is likely that they are strongly enriched.

To identify potential transcription factors binding to the enriched sequences, HOMER uses the TRANSFAC database. TRANSFAC contains experimentally-determined motifs for many transcription factors including various members of the Ets gene family. While the core Ets binding motif is similar among the various Ets family members, the flanking sequences are somewhat different. Using the TRANSFAC database, HOMER found that the best match to the enriched sequence was the Ets1 binding motif. However, we do agree that other related transcription factors in the Ets gene family may also bind to this sequence. For that reason, we have revised the text to include this point.

(6) I was confused by Fig. 8C. It appears here that several regions seem to be repressive in nature (ie less than promoter alone), so not sure why the results text says that none of the regions show strong repressive activity.

The reviewers is correct and we have revised the text accordingly.

(7) Given that GWAS signals suggest the importance of the Ets1 locus, do any of the SNPs map to putative regulatory elements highlighted in this study? If so what is their predicted effect on activity (ideally would also be tested but as a minimum discussed).

Several disease-associated SNPs do map close to or within putative enhancer regions. We’ve included a new figure showing where the SNPs associated with lupus map (new Supplemental Figure S5).

(8) Additional information for reviewer:

It should also be noted that some changes have been made to format the revised manuscript according to guidelines for PLoS One. As part of this effort, we have moved the figure legends into the main body of the text just under the paragraph in which they are first mentioned. This is consistent with the recommended formatting style for this journal - https://journals.plos.org/plosone/s/file?id=wjVg/PLOSOne_formatting_sample_main_body.pdf. The Supplemental Figure Legends are at the end of the manuscript and also included in the Supplemental figures themselves.

---

## [Decision Letter · Decision Letter 1]

22 Nov 2024

PONE-D-24-32251R1Sequences within and upstream of the mouse Ets1 gene drive high level expression in B cells, but are not sufficient for consistent expression in T cellsPLOS ONE

Dear Dr. Garrett-Sinha,

Thank you for re-submitting your manuscript to PLOS ONE. After further consideration, we feel that it has merit but it still does not fully meet PLOS ONE’s publication criteria as it currently stands. Therefore, we invite you to submit a revised version of the manuscript that addresses the points raised during the review process.

We look forward to receiving your revised manuscript.

Kind regards,

Roberto Mantovani

Academic Editor

PLOS ONE

**Journal Requirements:**

Reviewers' comments:

Reviewer's Responses to Questions

**Comments to the Author**

1. If the authors have adequately addressed your comments raised in a previous round of review and you feel that this manuscript is now acceptable for publication, you may indicate that here to bypass the “Comments to the Author” section, enter your conflict of interest statement in the “Confidential to Editor” section, and submit your "Accept" recommendation.

Reviewer #1: (No Response)

2. Is the manuscript technically sound, and do the data support the conclusions?

Reviewer #1: Yes

3. Has the statistical analysis been performed appropriately and rigorously? 

Reviewer #1: Yes

4. Have the authors made all data underlying the findings in their manuscript fully available?

Reviewer #1: Yes

5. Is the manuscript presented in an intelligible fashion and written in standard English?

Reviewer #1: Yes

6. Review Comments to the Author

**Reviewer #1: ** Most things have been addressed.

I appreciate this is PLOSONE and there are criteria for publication, and my initial comments were based on making the paper more conclusive rather than suggestive. I do take the arguments they make but it would be useful to add caveats about the limitations of the approach (ie surrounding previous points 1 and 2 about more appropriate definitive assays).

One this not addressed was "In Fig. 6C, the percentage of regions containing each motif should be added.

In the left table, the authors should comment on the second motif that resembles at

ETS motif. " The authors have not commented on the ETS-like motif. For the percentage of regions with motifs issue, then this is still useful to include as it provides a feel for what proportion of the effect is driven by a particular TF. You can get highly enriched regions in a low percentage of regions if the genomic back ground is also low for example. Again, this is not essential but just helps the reader to gauge the influence of ETS TFs.

new Supplemental Figure S5 was not included or referred to in the revised manuscript so was impossible to evaluate.

7. PLOS authors have the option to publish the peer review history of their article (what does this mean? ). If published, this will include your full peer review and any attached files.

**Do you want your identity to be public for this peer review?** For information about this choice, including consent withdrawal, please see our Privacy Policy .

Reviewer #1: No

---

## [Author Response · Author response to Decision Letter 2]

24 Nov 2024

Response to Reviewer Comments

Reviewer #1: Most things have been addressed.

I appreciate this is PLOSONE and there are criteria for publication, and my initial comments were based on making the paper more conclusive rather than suggestive. I do take the arguments they make but it would be useful to add caveats about the limitations of the approach (ie surrounding previous points 1 and 2 about more appropriate definitive assays).

We have added some sentences to address this on pages 22-23 of the revised paper.

One this not addressed was "In Fig. 6C, the percentage of regions containing each motif should be added.

While we would like to address this point, unfortunately the software used to identify motifs (HOMER software - http://homer.ucsd.edu/homer/) does not provide the percentage of sequences that contain a motif. It simply identifies motifs that are over-represented in one dataset (e.g., stimulated B cells) as compared to the other dataset (e.g., unstimulated B cells) and gives a p value for the significance of the over-representation. The p values for the motifs we list are all very small, indicating that these sequences are highly enriched in one dataset or the other. But we cannot provide a percentage of sites that contain those particular motifs.

In the left table, the authors should comment on the second motif that resembles at

ETS motif. " The authors have not commented on the ETS-like motif. For the percentage of regions with motifs issue, then this is still useful to include as it provides a feel for what proportion of the effect is driven by a particular TF. You can get highly enriched regions in a low percentage of regions if the genomic back ground is also low for example. Again, this is not essential but just helps the reader to gauge the influence of ETS TFs.

We agree with the reviewer’s point, but unfortunately as described above, we have no information on the percentages of sequences with a particular motif.

new Supplemental Figure S5 was not included or referred to in the revised manuscript so was impossible to evaluate.

Sorry, this was a mistake on our part. The extra figure is Supplemental Figure 4, which was included in the figures along with a figure legend, but was not referenced in the text. We have added sentences on page 14 of the manuscript to refer to the new figure.

---

## [Editor Report · Decision Letter 2]

12 Dec 2024

Sequences within and upstream of the mouse Ets1 gene drive high level expression in B cells, but are not sufficient for consistent expression in T cells

PONE-D-24-32251R2

Dear Dr. Garrett-Sinha,

We’re pleased to inform you that your manuscript has been judged scientifically suitable for publication and will be formally accepted for publication once it meets all outstanding technical requirements.

Kind regards,

Roberto Mantovani

Academic Editor

PLOS ONE
---

## [Editor Report · Acceptance letter]

PONE-D-24-32251R2

PLOS ONE

Dear Dr. Garrett-Sinha,

I'm pleased to inform you that your manuscript has been deemed suitable for publication in PLOS ONE. Congratulations! Your manuscript is now being handed over to our production team.

Kind regards,

on behalf of

Prof. Roberto Mantovani

Academic Editor

PLOS ONE